# Multi-Dimensional Constraint Integration Method for Large Language Models via Lyapunov Stability Theory

## Abstract

Large language model agents face a fundamental challenge of environmental understanding deficiency in multi-constraint environments: they rely on textual pattern matching rather than deep environmental modeling, leading to decisions that are disconnected from environmental requirements. The root cause lies in the general pre-training paradigm's lack of constraint-structured annotations, causing models to treat multiple constraints as independent fragments without capturing inter-constraint dependencies and environmental dynamic characteristics. This paper proposes a Lyapunov-guided multi-constraint aware decoding framework that innovatively adapts Lyapunov stability theory to discrete language generation processes. By constructing a multi-constraint Lyapunov modeling system, constraint deviations are quantified as Lyapunov functions, enabling agents to quantitatively assess constraint satisfaction distances and obtain optimal convergence directions. Experimental validation demonstrates that this method significantly improves constraint satisfaction rates while maintaining generation quality.

## 1 Introduction

Current large language model (LLM) agents face significant challenges in multi-constraint understanding when making decisions in complex environments. Existing agents primarily rely on text pattern matching and statistical inference, lacking deep environmental comprehension capabilities Zou et al. (2025). This limitation prevents them from generating decisions that align with environmental characteristics in multi-constraint scenarios, resulting in agent behaviors that disconnect from actual environmental requirements and severely impact task execution effectiveness show in Fig 1.

The root cause lies in the inherent limitations of LLM pre-training paradigms. While LLMs based on general text pre-training cover extensive knowledge, they lack structured annotation for specific environmental constraints and tend to treat multiple constraints as independent text fragments Wang et al. (2024); Xie et al. (2025). This approach fails to capture inter-constraint relationships and overall environmental characteristics. Compared to specialized environmental modeling methods, LLMs rely on surface-level textual features for reasoning, lacking deep understanding of constraint hierarchical structures and dynamic changes, leading to improper constraint conflict handling and poor environmental adaptability Liu et al. (2024b).

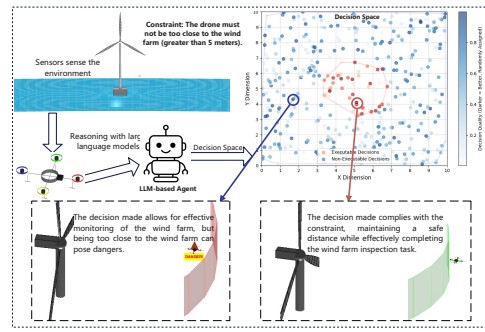

Figure 1: LLM agents generate reasonable decisions that often violate environmental constraints, creating a critical decision-execution gap in real-world applications.

The core challenges of multi-constraint environment understanding center on the absence of unified measurement mechanisms and convergence guarantees. Heterogeneous constraint types are

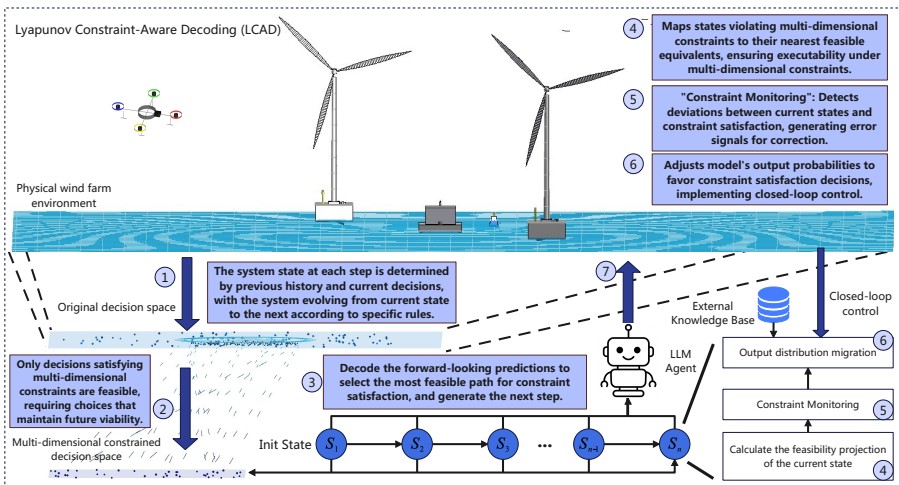

Figure 2: Overview of the Lyapunov Constraint-Aware Decoding (LCAD) framework applied to UAV wind farm inspection.

difficult to compare and optimize within the same mathematical space Ming et al. (2024). Dynamic coupling between constraints causes local satisfaction to trigger global conflicts, trapping agents in cyclic adjustments without achieving global optima. Existing methods generally lack theoretical convergence guarantees, causing agents to oscillate at constraint boundaries or converge to local sub-optimal solutions, while lacking predictive capabilities for constraint evolution trends Chow et al. (2018).

Lyapunov stability theory provides a theoretical foundation for addressing this problem. Richards et al. (2018) The Lyapunov function serves as a "deviation metric," not only quantifying the distance between the system and ideal states but also indicating the optimal path toward stability through gradient information Verma et al. (2025). Its 'progressive improvement' paradigm requires each step to be closer to the target than the previous one, ensuring convergence through monotonic improvement, which naturally aligns with the autoregressive characteristics of LLMs. This unified framework can transform multi-dimensional constraint fusion problems into single 'system deviation' optimization, avoiding coordination difficulties when handling constraints separately Hejase & Ozguner (2023). However, traditional Lyapunov theory was designed for continuous dynamical systems and has not been applied to large language model agents.

Therefore, based on Lyapunov stability theory, we construct a Lyapunov modeling system for multi-constraint environments (Fig 2). By quantifying constraint deviations as Lyapunov functions, LLM agents can assess distances to constraint satisfaction states and obtain optimal convergence directions. We develop the Lyapunov Constraint-Aware Decoding (LCAD) algorithm, mapping discrete text generation to continuous constraint optimization for controllable agent decision-making in complex multi-constraint environments.

Our main contribution is innovatively adapting Lyapunov stability theory to the discrete stochastic generation process of large language models and proposing the Lyapunov Constraint-Aware Decoding (LCAD) algorithm framework. Experiments show that our method improves constraint satisfaction rates by 5.5% compared to baseline methods while maintaining text fluency and semantic consistency.

## 2 RELATED WORK

### 2.1 LLM-BASED INTELLIGENT SYSTEMS

Large Language Model (LLM)-based intelligent systems have advanced natural language processing and decision-making, enabling applications in virtual assistants and multi-agent systems Gao et al. (2024). However, their environmental understanding remains limited, particularly in dynamic set-

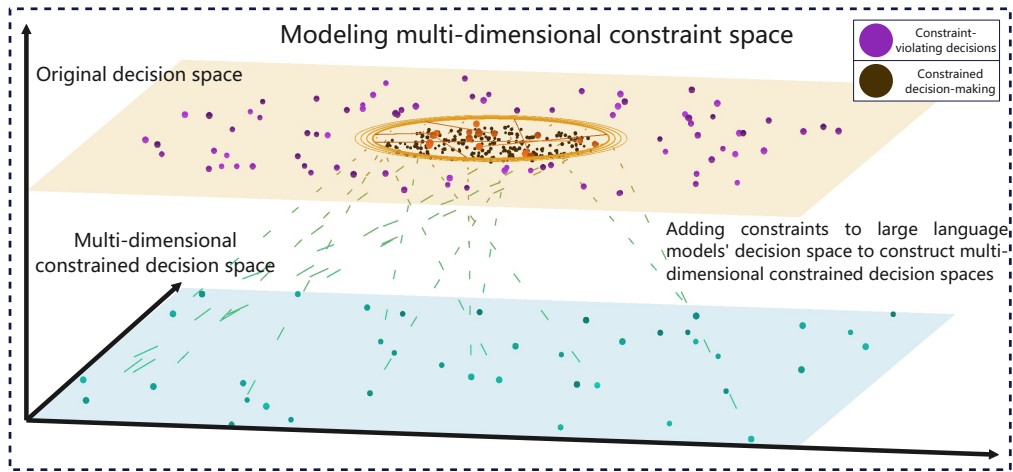

Figure 3: Lyapunov modeling constrains decision space, focusing feasible solutions within boundaries.

tings. LLM-based agents struggle to transition from virtual to physical environments due to reliance on textual abstractions that fail to capture multimodal inputs Bigazzi et al. (2023). Benchmarks like ALFWorld show agents excel in text-based tasks but falter in real-world scenarios requiring environmental adaptation Shridhar et al. (2020), while WebArena reveals poor generalization to unseen contexts Zhou et al. (2023). Hallucinations exacerbate these issues by generating incorrect environmental interpretations Ji et al. (2023), and LLMs exhibit weak robustness against adversarial inputs Zhu et al. (2024). Limited long-tail knowledge hinders handling of domain-specific contexts Liu et al. (2024a), while multi-agent systems face challenges in collective environmental understanding Perry et al. (2025). Integrating multimodal inputs for comprehensive environmental perception remains problematic Wu et al. (2023), highlighting the need for enhanced environmental modeling.

## 2.2 MULTI-BEAM INCORPORATION METHODS FOR LARGE LANGUAGE MODELS

Constraint incorporation in large language models encompasses four primary categories. Decoding constraint methods modify search algorithms, including grid beam search Hokamp & Liu (2017) and neural logic decoding, but suffer from high computational complexity and may impair fluency. Fine-tuning strategies incorporate control codes into pre-trained models, such as CTRL Keskar et al. (2019) and DExperts Liu et al. (2021), but struggle with complex constraints Lu et al. (2021). External guidance methods employ classifiers like PPLM Pascual et al. (2021), offering flexibility but incurring high costs and lacking unified metrics. Editing prototype methods achieve constraint incorporation through iterative modification while maintaining semantic coherence. Existing approaches remain limited by model expressiveness and constraint optimization challenges Anderson et al. (2017), where non-differentiable optimization yields sparse signals Choshen et al. (2019) and absent evaluation metrics hinder assessment van der Lee et al. (2019).

## 3 METHODOLOGY

This section establishes theoretical foundations for environment-aware LLM decision generation by defining state space and reachability in constraint environments through control theory perspectives (Figure 3).

Consider the state of an LLM agent in a multi-constraint environment as $s_t = (y_{<t}, y_t, C_t)$, where $y_{<t}$ denotes the generation history, $y_t$ represents the current output, and $C_t$ characterizes the dynamic constraint environment. The feasible region of the multi-constraint environment is defined as the subset of states satisfying all environmental constraints:

$$\mathcal{F} = \{s_t \in \mathcal{S} \,|g_i(s_t, C_t) \leq 0,$$
$$i = 1, 2, \ldots, m\} \tag{1}$$

where $g_i(s_t, C_t)$ represents the $i$-th environmental constraint function and $m$ denotes the total number of constraints.

From a dynamical systems perspective, agent state evolution is formalized as $s_{t+1} = f(s_t, y_t, C_t)$, where $f$ is the state transition function incorporating environmental dynamics. In multi-constraint environments, $y_t$ serves as the system's control input, requiring state transitions to maintain constraint satisfaction. We define the environment-aware feasible control input set:

$$\mathcal{U}_t(C_t) = \{y_t \,|s_{t+1} = f(s_t, y_t, C_t)$$
$$\in \mathcal{F}\} \tag{2}$$

The core challenge in multi-constraint environments lies in complex inter-constraint coupling relationships. Traditional approaches often treat different constraints independently, neglecting mutual dependencies. We define the composite distance from agent state to the multi-constraint feasible region:

$$d(s_t, \mathcal{F}) = \sqrt{\sum_{i=1}^{m} w_i \cdot \max(0, g_i(s_t, C_t))^2} \tag{3}$$

where the weights $w_i(C_t) = \alpha_i \cdot \pi_i(C_t) + \beta_i \cdot \phi_i(s_t, C_t)$ are dynamically adjusted, with $\pi_i(C_t)$ and $\phi_i(s_t, C_t)$ representing environmental priority and state-dependent urgency of constraints, respectively.

To address inter-constraint dependencies, we introduce a constraint dependency graph characterizing these complex relationships through a dependency strength matrix $D$:

$$D_{ij} = \begin{cases} \rho_{ij} & \text{if } (C_i, C_j) \in E \\ 0 & \text{otherwise} \end{cases} \tag{4}$$

The coupling-aware distance function more accurately reflects the agent's true deviation from the feasible region.

From Lyapunov stability theory, modeling the multi-constraint environment feasible region is equivalent to defining the agent's constraint harmony region. We construct a multi-constraint Lyapunov function:

$$V(s_t, C_t) = d_{\text{coupled}}(s_t, \mathcal{F})^2$$
$$+ \gamma \sum_{i<j} \xi_{ij}(s_t, C_t) \tag{5}$$

where $\gamma > 0$ is the conflict penalty weight and $\xi_{ij}(s_t, C_t)$ quantifies inter-constraint conflict intensity. When the agent state satisfies all constraints without conflicts, $V(s_t, C_t) = 0$; otherwise, the function assumes positive values, providing clear optimization direction.

The Lyapunov stability condition requires:

$$\dot{V}(s_t, C_t) = \nabla_{s_t} V \cdot \dot{s}_t + \nabla_{C_t} V \cdot \dot{C}_t \leq 0 \tag{6}$$

This condition ensures state convergence toward the feasible region. To handle dynamic evolution of environmental constraints, the time-varying characteristics of constraints must satisfy:

$$\frac{d}{dt} g_i(s_t, C_t) = \nabla_{s_t} g_i^T \dot{s}_t + \nabla_{C_t} g_i^T \dot{C}_t \leq 0 \tag{7}$$

To ensure long-term stability in dynamic environments, we define the reachable set:

$$\mathcal{R}_T(s_0, C_0) = \{s_T \mid \exists y_{0:T-1}, s_{t+1} = f(s_t, y_t, C_t),$$
$$s_t \in \mathcal{F}, \forall t \in [0, T-1]\} \tag{8}$$

This framework unifies heterogeneous constraints into a single Lyapunov function, transforming multi-constraint satisfaction into minimization optimization while establishing a rigorous foundation for control design.

---

**Algorithm 1:** Multi-Constraint Aware Predictive Decoding

---

**Input:** Initial state $s_0$, constraints $C_0$, horizon $H$, weight $\lambda$
**Output:** Generated sequence $Y$
**while** *not end of generation* **do**
    // Predict H-step future sequences
    $\mathcal{Y} \leftarrow \text{SampleFutureSequences}(s_t, H)$;
    **foreach** *sequence $Y_{t:t+H} \in \mathcal{Y}$* **do**
        $score_{llm} \leftarrow \sum_{k=t}^{t+H} \log P(y_k | y_{<k})$;
        $violations \leftarrow 0$;
        **for** $k = t$ **to** $t + H$ **do**
            **for** $i = 1$ **to** $m$ **do**
                $violations \leftarrow violations + w_i \cdot \max(0, g_i(y_k, C_k))^2$;
            // Add coupling penalties
            **for** *each constraint pair $(i, j)$* **do**
                $violations \leftarrow violations + \omega_{ij} \cdot \max(0, g_i) \cdot \max(0, g_j)$;
        $J(Y_{t:t+H}) \leftarrow score_{llm} - \lambda \cdot violations$;
    // Select optimal and execute first token
    $Y_{t:t+H}^* \leftarrow \arg\max_Y J(Y)$;
    $\text{Execute}(Y_t^*)$;
    // Update state with receding horizon
    $s_{t+1} \leftarrow f(s_t, Y_t^*, C_t)$;
    $C_{t+1} \leftarrow h(C_t, s_t, Y_t^*)$;
    // Adaptive weight update
    **for** $i = 1$ **to** $m$ **do**
        $w_i \leftarrow w_i + \eta_w \cdot \nabla_{w_i} \mathcal{L}_{\text{violation}}$;
    $t \leftarrow t + 1$;

---

### 3.1 MULTI-CONSTRAINT AWARE PREDICTIVE DECODING FRAMEWORK

We propose an MPC-based framework that predicts future states over horizon $H$ to optimize LLM outputs while ensuring long-term constraint satisfaction. At step $t$, given the current state $s_t = (y_{<t}, y_t, C_t)$ and multi-constraint feasible region $\mathcal{F}$, we optimize:

$$J(Y_{t:t+H}) = \sum_{k=t}^{t+H} [\log P(y_k | y_{<k}, X) - \lambda \cdot \Phi(g_1, \ldots, g_m)] \tag{9}$$

where $\Phi$ captures both individual constraint violations and their interactions:

$$\Phi(g_1, \ldots, g_m) = \sum_{i=1}^{m} w_i \max(0, g_i)^2 + \sum_{i<j} \omega_{ij} \max(0, g_i) \max(0, g_j) \tag{10}$$

The multi-constraint aware optimization employs a receding horizon strategy that executes only the first optimal action $y_t^*$, then updates the state $s_{t+1} = f(s_t, y_t^*, C_t)$ and constraints $C_{t+1} = h(C_t, s_t, y_t^*)$ before replanning. Algorithm 1 presents the complete decoding process.

The receding horizon strategy balances computational efficiency with long-term planning. When $H = 1$, this reduces to greedy decoding; larger $H$ improves constraint awareness at increased computational cost. The adaptive weight updates ensure the system learns from constraint violations to improve future performance.

### 3.2 CONSTRAINT DEVIATION MONITORING AND MULTI-DIMENSIONAL PROJECTION MECHANISM

When states deviate from the feasible region $\mathcal{F}$, we employ real-time monitoring and projection to ensure constraint satisfaction. This mechanism complements predictive decoding by providing immediate corrections.

We monitor deviations using the composite distance considering constraint interactions:

$$d(s_t, \mathcal{F}) = \sqrt{\sum_{i=1}^{m} w_i \cdot \max(0, g_i(s_t, C_t))^2 + \sum_{i<j} \omega_{ij} \cdot h_{ij}} \tag{11}$$

where $h_{ij} = \max(0, g_i(s_t, C_t)) \cdot \max(0, g_j(s_t, C_t))$ captures coupling between constraints.

Algorithm 2 presents the multi-dimensional projection mechanism that maps infeasible states back to the feasible region through coordinated gradient adjustments.

The projection ensures the Lyapunov stability condition $\dot{V}(s_t, C_t) = 2d(s_t, \mathcal{F}) \cdot \nabla d^T \dot{s}_t \leq 0$, guaranteeing convergence to the feasible region. The corrected probability distribution reduces the likelihood of constraint-violating outputs through exponential penalties.

### 3.3 ENHANCED CONTROLLABILITY STRATEGY WITH EXTERNAL KNOWLEDGE

This section presents an external knowledge-enhanced strategy to adjust LLM outputs and adapt to multi-constraint feasible regions $\mathcal{F}$, addressing data-driven method limitations through environmental knowledge injection and adaptive feedback control.

We define environment-aware controllability as the reachability from initial state $s_0$ to target state $s_T \in \mathcal{F}$ under dynamic environment $C_{0:T}$, where there exists a control input sequence $u_{0:T-1}$ and environmental adaptation strategy $\pi$ satisfying the state transition:

$$s_{t+1} = f(s_t, u_t, C_t), \quad s_T \in \mathcal{F}, \quad u_t = \pi(s_t, C_t, \mathcal{K}_t) \tag{12}$$

where $u_t$ represents the environment-aware external regulation signal and $\mathcal{K}_t$ denotes the external knowledge base at time $t$. The environment-aware control set is:

$$\mathcal{U}_t(C_t) = \{u_t \,|\, f(s_t, u_t, C_t) \in \mathcal{F} \wedge \text{compatible}(u_t, C_t)\} \tag{13}$$

Environmental knowledge injection is realized through multi-layer conditional probability adjustment. By incorporating multi-constraint environment $\mathcal{C}_t$ and external knowledge $\mathcal{K}_t$ into the probability distribution, we modify $P(y_t|y_{<t}, X)$ to:

---

**Algorithm 2:** Multi-Dimensional Constraint Projection

---

**Input:** State $s_t$, constraints $\{g_1, \ldots, g_m\}$, weights $\{w_1, \ldots, w_m\}$
**Output:** Projected state $s_t^*$ and corrected distribution $P'(y_t)$
// Detect constraint violations
**for** $i = 1$ **to** $m$ **do**
$\quad \lfloor \ v_i \leftarrow \max(0, g_i(s_t, C_t))$;
**if** $\sum_i v_i = 0$ **then**
$\quad \lfloor$ **return** $s_t, P(y_t | y_{<t})$ // Already feasible
// Compute multi-constraint gradients
**foreach** *constraint i* **do**
$\quad \nabla_i \leftarrow \nabla_{s_t} g_i(s_t, C_t)$;
$\quad$ // Consider coupling with other constraints
$\quad$ **foreach** *coupled constraint* $j \in \mathcal{N}(i)$ **do**
$\quad \quad \lfloor \ \nabla_i \leftarrow \nabla_i + \omega_{ij} \cdot \nabla_{s_t}[g_i \cdot g_j]$;
// Project via gradient descent
$s_t^* \leftarrow s_t$;
**for** $iter = 1$ **to** $MaxIter$ **do**
$\quad$ **foreach** *constraint i with* $v_i > 0$ **do**
$\quad \quad \lfloor \ s_t^* \leftarrow s_t^* - \eta_i \cdot w_i \cdot \nabla_i$;
$\quad$ **if** $d(s_t^*, \mathcal{F}) < \epsilon$ **then**
$\quad \quad \lfloor$ **break**;
// Adjust output probability distribution
$P'(y_t) \leftarrow P(y_t | y_{<t}) \cdot \exp\left(-\sum_i \alpha_i v_i - \sum_{i<j} \alpha_{ij} v_i v_j\right)$;
$P'(y_t) \leftarrow \text{Normalize}(P'(y_t))$;
**return** $s_t^*, P'(y_t)$;

---

$$P(y_t | y_{<t}, X, \mathcal{C}_t, \mathcal{K}_t) =$$
$$\frac{P(y_t | y_{<t}, X) \cdot \exp\left(-\sum_{i=1}^m \beta_i d_i - \zeta \cdot \text{semantic\_conflict}(y_t, \mathcal{K}_t)\right)}{\mathcal{Z}_t} \quad (14)$$

where $\beta_i > 0$ are regulation parameters for the $i$-th constraint, $\zeta > 0$ is the semantic conflict weight, and semantic_conflict measures the semantic conflict degree between output and external knowledge. This distribution reduces probabilities of constraint-violating and semantically inconsistent outputs through exponential decay.

In practice, $\mathcal{C}_t$ and $\mathcal{K}_t$ can be introduced through hierarchical retrieval-augmented generation or dynamic prompt engineering. We define the environment-aware knowledge injection strategy:

$$u_t = \text{compose}(\text{constraint\_prompt}(C_t), \text{knowledge\_prompt}(\mathcal{K}_t),$$
$$\text{adaptive\_rule}(s_t, e_t)) \quad (15)$$

The environment-aware feedback mechanism further optimizes controllability through multi-dimensional closed-loop control. We define the multi-constraint error signal vector:

$$\mathbf{e}_t = [e_1^{(t)}, e_2^{(t)}, \ldots, e_m^{(t)}]^T, \quad e_i^{(t)} = \max(0, g_i(s_t, C_t)) \quad (16)$$

Multi-dimensional feedback adjustment is triggered when $\|\mathbf{e}_t\|_2 > 0$. The environment-aware state update becomes:

$$s_{t+1} = s_t - \sum_{i=1}^{m} \eta_i \nabla_{s_t} d_i(s_t, \mathcal{F}_i) - \xi \nabla_{s_t} \text{env\_misalign}(s_t, C_t) \tag{17}$$

where $\text{env\_misalign}(s_t, C_t)$ measures the misalignment degree between state and environment.

To ensure stability in multi-constraint environments, we extend the Lyapunov function to an environment-aware form:

$$V(s_t, C_t) = \frac{1}{2} \sum_{i=1}^{m} w_i d_i(s_t, \mathcal{F}_i)^2 + \frac{1}{2} \rho \cdot \text{env\_misalign}(s_t, C_t)^2 \tag{18}$$

This negative definiteness guarantees convergence of $s_t$ to the multi-constraint feasible region under bounded environmental drift conditions.

Considering environment-aware state transitions, the control input optimization becomes:

$$u_t^* = \arg \min_{u_t} \Big[ \sum_{i=1}^{m} \alpha_i ||f_i(s_t, u_t, C_t) - s_{t,i}^*||_2^2$$
$$+ \delta \cdot \text{knowledge\_conflict}(u_t, \mathcal{K}_t) \Big] \tag{19}$$

## 4 EXPERIMENTS

### 4.1 EXPERIMENTAL SETUP

We conduct experiments on Microsoft AirSim Shah et al. (2018) using LLaMA-3 8B Grattafiori et al. (2024) with temperature 0.3 and 256-token limit for real-time control.

The scenario involves offshore wind farm inspection with constraints including turbine blade safety distance ($\geq$15m), altitude ($\geq$20m), velocity limits, inspection coverage, and battery endurance. These heterogeneous constraints exhibit complex coupling relationships.

Evaluation uses two metrics: constraint satisfaction rate (percentage of time steps satisfying all constraints) and flight crash rate (collision frequency from constraint violations).

### 4.2 EXPERIMENTAL RESULT

We evaluate four representative constraint handling methods. Chain-of-Thought Wei et al. (2022) enhances constraint understanding through step-wise reasoning but suffers breakdown under multi-constraint conflicts. Tree-of-Thoughts Yao et al. (2023) uses tree-based search for multiple reasoning paths with high computational cost and limited adaptability.

Table 1: Overall Performance Metrics Comparison

| Method | Constraint Sat. (%) | | Collision Rate (%) | |
|---|---|---|---|---|
| | Rate | Task | Rate | Avg |
| Chain-of-Thought | 58.4±2.6 | 54.2±3.1 | 19.8±2.2 | 58.4 |
| Tree-of-Thoughts | 62.1±2.4 | 58.7±2.8 | 18.3±1.9 | 62.1 |
| DExperts | 63.8±2.7 | 61.2±3.1 | 17.4±1.9 | 63.8 |
| PPLM | 65.4±2.3 | 63.7±2.8 | 16.9±1.6 | 65.4 |
| **LCAD (Ours)** | **71.2±2.1** | **68.5±2.5** | **13.8±1.4** | **71.2** |

DExperts Liu et al. (2021) employs mixture-of-experts for specialized constraint models, excelling on specific constraints but lacking generalization. PPLM Dathathri et al. (2019) uses plug-and-play architecture with external classifiers for probability adjustment, offering flexibility but limited

Table 2: Individual Constraint Satisfaction Comparison (%)

| Method | Blade | Alt. | Vel. | Task | Batt. |
|---|---|---|---|---|---|
| Chain-of-Thought | 55.2 | 62.1 | 57.3 | 54.2 | 60.8 |
| Tree-of-Thoughts | 59.7 | 65.8 | 61.4 | 58.7 | 64.2 |
| DExperts | 61.2 | 67.9 | 62.4 | 61.2 | 66.8 |
| PPLM | 63.1 | 69.7 | 64.6 | 63.7 | 68.2 |
| **LCAD (Ours)** | **68.9** | **74.8** | **70.3** | **68.5** | **73.6** |

multi-constraint coordination. These methods work well for single constraints but lack unified multi-constraint frameworks and theoretical convergence guarantees.

As show in Table 1, LCAD achieves optimal performance across all metrics. Compared to PPLM, LCAD improves constraint satisfaction by 5.8%, reduces collision rate by 3.1%, and enhances task completion by 4.8% (Fig **??**). These improvements are significant for safety-critical UAV applications, with collision reduction demonstrating effective Lyapunov-guided safety enhancement.

In Table 2, LCAD demonstrates consistent advantages across all constraints. Safety-critical improvements are pronounced: blade safety increases by 5.8% and altitude by 5.1%. This balanced improvement reflects effective heterogeneous constraint integration, avoiding traditional methods' constraint neglect or over-optimization. Battery constraint improvement of 5.4% shows LCAD balances safety and efficiency effectively.

## 5 DISCUSSION

The intelligent agent decision-making problem in multi-constraint environments presents profound complexity due to the absence of unified constraint measurement mechanisms and convergence guarantees. The discrete stochastic generation process of LLMs inherently lacks the continuity characteristics required by traditional constraint handling methods.

Our approach addresses this challenge by adapting Lyapunov stability theory's "distance measurement, directional guidance, and progressive improvement" paradigm to discrete stochastic language generation. The Lyapunov function unifies heterogeneous constraints into a single deviation metric, transforming the complex multi-constraint satisfaction problem into a minimization optimization where each generation step advances toward constraint satisfaction. This enables agents to evaluate constraint deviations in real-time and obtain explicit improvement directions, ensuring theoretical convergence guarantees.

Experimental results validate our theoretical framework. LCAD achieves 71.2% constraint satisfaction rate, outperforming PPLM by 5.8%, with a 3.1% reduction in collision rate—critical improvements for safety-critical applications. The balanced performance across all constraint types (Table 1) demonstrates effective heterogeneous constraint integration without the constraint neglect or over-optimization issues of traditional methods.

This work establishes systematic connections between control theory and natural language processing, providing theoretical guarantees for AI system trustworthiness. The unified framework reduces deployment barriers in complex environments while enabling flexible configuration across scenarios. However, extending to multi-agent environments, where individual and global constraints must be balanced, remains an important future direction.

## 6 CONCLUSION

This work addresses inadequate environmental comprehension in large language model agents within multi-constraint environments by proposing a Lyapunov stability theory-grounded framework. The approach quantifies heterogeneous constraints as Lyapunov functions, enabling agents to assess constraint deviations and obtain convergence guidance. The Lyapunov Constraint-Aware Decoding (LCAD) algorithm integrates predictive lookahead, multi-dimensional projection, and environment-aware enhancement to map discrete text generation to continuous constraint optimization.

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
