# OpenReview forum: "Multi-Dimensional Constraint Integration Method for Large Language Models via Lyapunov Stability Theory"
_ICLR.cc/2026/Conference — ICLR 2026 Conference Withdrawn Submission_

### Official Review · Reviewer_Y9ju · 2025-10-15

**Soundness:** 2
**Presentation:** 2
**Contribution:** 2
**Rating:** 2
**Confidence:** 3

**Summary:**

This paper proposes a Lyapunov-guided multi-constraint aware decoding framework that allows agents to quantitatively assess constraint satisfaction distances and obtain optimal convergence directions.

**Strengths:**

1. This paper targets a useful and pratical application.
2. Easy to understand and follow.
3. Detailed theory is discussed.

**Weaknesses:**

1. There is a lack of a sufficient literature review. In the introduction part, there is only one sentence: "Existing methods generally lack theoretical convergence guarantees, causing agents to oscillate at constraint boundaries or converge to local sub-optimal solutions, while lacking predictive capabilities for constraint evolution trends", making the motivation not convincing enough. Besides, I notice that the discussed existing works are no later than 2021, lacking advanced works. Comparing with such old baselines makes the effectiveness of the method not strong.
2. The experiments are not solid. Only one benchmark and base model is used. The setting of a 256-token limit is not normal, needing an explanation for it. Also, the baselines are weak and old. Comparison with recent works is essential to demonstrate the advantages. More important experiments such as computational cost are also need to be conducted.
3. Writing mistakes. On page 9, there is Fig ??.
4. Implementation details are unclear. The approach requires gradients such as $\nabla g_i$, but the state includes discrete actions. The source of these gradients and any approximation method are not specified.
5. The discussion part is meaningless. It seems like a summarization of the proposed method and experimental results without any additional information. A discussion part may introduce the underlying driver of the performance and potential insights.

**Questions:**

See weaknesses section.

---

### Official Review · Reviewer_2YSn · 2025-11-01

**Soundness:** 2
**Presentation:** 2
**Contribution:** 4
**Rating:** 4
**Confidence:** 3

**Summary:**

The paper innovatively adapts Lyapunov Stability Theory from control of continuous dynamical systems to the discrete text-generation process of large language models (LLMs), proposing a Lyapunov Constraint-Aware Decoding (LCAD) framework to address the core issue of misalignment between multi-constraint decision-making and environmental requirements in LLM agents. It targets the limitations of pattern-matching behavior in current LLMs and the lack of structured modeling of constraints and dynamic couplings, offering a cross-disciplinary solution. The authors validate LCAD on a safety-critical drone inspection task in a wind farm and report superior constraint satisfaction and collision avoidance rates. The work is theoretically novel and practically relevant. However, there are shortcomings in the completeness of theoretical details, the adequacy of experimental design, and the depth of result analysis.

**Strengths:**

1. Notable theoretical contribution. To the best of my knowledge, this is the first attempt to transplant Lyapunov stability—originally formulated for continuous-time dynamical systems—to the discrete stochastic space of LLM generation. The idea supplies a unified optimization objective and a formal guarantee for multi-constraint problems, and it carries clear physical and control-theoretic meaning.
2. Well-structured technical framework. LCAD integrates predictive decoding, a multi-dimensional projection mechanism, and external knowledge augmentation into a closed-loop "predict–monitor–correct–feedback" control logic that is operationally practical for complex multi-constraint environments.
3. Strong scenario specificity. UAV wind-farm inspection involves multi-dimensional, heterogeneous, strongly coupled constraints and is representative of safety-critical applications. The method outperforms baselines in constraint satisfaction and collision rates, suggesting practical potential.

**Weaknesses:**

1. Figure design and citation formatting. Figures contain lengthy sentences and dense text, which is inconsistent with schematic clarity and readability. In Figure 1 the word "large" is partly occluded.  The manuscript mixes "Figure" and "Fig" in citations, and there is a reference error in the experiments (around line 446).
2. Symbol reuse and incomplete definitions. Equations 3 and 11, 2 and 13, 5 and 18 reuse identical symbols for different quantities. In Equation 4, the computation of the dependency-strength matrix is unspecified; does $\rho$ have the same meaning as in Equation 18? In Equation 5, $\xi$ is defined as a matrix of inter-constraint conflict strengths, yet in Equation 17, $\xi$ appears as a hyperparameter without clear definition. The $\alpha$ and $\beta$ at line 188, the $\alpha$ in Algorithm 2 and Equation 19, and the $\beta$ in Equation 14 denote different weights—these should be distinguished with consistent notation. A thorough pass to ensure consistency and rigor in symbol definitions is recommended.
3. Insufficient justification for Lyapunov-function suitability. The paper heuristically maps constraint deviation to a Lyapunov function but does not establish how the discrete text-generation process satisfies the core assumptions of Lyapunov stability theory, nor does it provide a rigorous convergence proof in the discrete language generation space.
4. Incomplete experimental analysis. Given the real-time nature of UAV inspection, please report computational overhead and efficiency for multi-step prediction, constraint-coupling modeling, and gradient projection. Add comparisons of decision-making trajectories between LCAD and baselines to illustrate how Lyapunov gradients guide globally better choices, and include ablations to quantify each module's contribution, with deeper analysis of results.
5. Applicability and generalization. Experiments are limited to a single scenario. It remains unclear how the approach would handle constraints that are more open-ended, semantically complex, or subjective. Since the paper claims to preserve fluency and semantic consistency, evaluations in other constrained text-generation settings should be added, using standard quality metrics for comparison against baselines.

**Questions:**

1. In Table 1, one metric under flight conflict rate is labeled "Avg" but its meaning is not defined in the text. It shows an increasing trend and is identical to the first column's constraint satisfaction rate. Is this a mistake?
2. The paper provides only abstract definitions for the constraint function, constraint dependency strength, env_misalign, and semantic/knowledge_conflict, without concrete functional forms. In non-physical settings with unstructured constraints, how should these functions be quantified?
### Rating

---

### Official Review · Reviewer_fjEF · 2025-11-01

**Soundness:** 3
**Presentation:** 3
**Contribution:** 3
**Rating:** 6
**Confidence:** 4

**Summary:**

This paper focuses on the insufficient environmental understanding of large language model (LLM) agents in multi-constraint environments. It identifies the root cause as the lack of constraint-structured annotations in LLM pre-training, which makes models treat multiple constraints as independent fragments and fail to capture inter-constraint dependencies and environmental dynamics.

The paper adapts Lyapunov stability theory (originally for continuous dynamical systems) to the discrete generation process of LLMs, proposing the Lyapunov Constraint-Aware Decoding (LCAD) framework. This framework quantifies constraint deviations as Lyapunov functions, and integrates three modules—model predictive control (MPC)-based predictive decoding, multi-dimensional constraint projection, and external knowledge enhancement—to form a closed loop, realizing the mapping between text generation and constraint optimization.

Experiments were conducted on the Microsoft AirSim platform using the LLaMA-3 8B model. Compared with four baseline methods, the framework achieved improvements across all single constraint dimensions while maintaining text quality. The work establishes a connection between control theory and natural language processing, provides theoretical guarantees for AI reliability, and reduces the threshold for deployment in complex environments.

**Strengths:**

1. It is the first work to adapt Lyapunov stability theory to the discrete generation process of LLMs, addressing two core challenges in multi-constraint environments. By converting multi-dimensional heterogeneous constraints into a single "system deviation" optimization objective, it avoids the fragmentation issues of traditional constraint-handling methods. Leveraging the "progressive improvement" paradigm, it provides strict mathematical convergence guarantees for constraint satisfaction, filling the gap of insufficient theoretical support in traditional methods.

2. The LCAD framework covers the entire lifecycle of agent decision-making and forms a closed-loop control. Before decision-making, MPC-based predictive decoding optimizes future sequences over H steps to balance long-term constraint satisfaction and computational efficiency. During decision-making, the multi-dimensional projection mechanism detects constraint deviations in real time and maps infeasible states back to the feasible region, preventing deviation accumulation. After decision-making, external knowledge injection and adaptive weight updates continuously optimize subsequent generations, effectively avoiding the limitations of single-module designs and ensuring sustained constraint satisfaction.

3. Experiments were designed for the safety-critical scenario of UAV wind farm inspection, focusing on five high-priority constraints. Quantitative results show significant improvements in safety-related constraints, demonstrating the practical value of the method in high-risk real-world applications. Meanwhile, by comparing with four representative baseline methods (Chain-of-Thought, Tree-of-Thoughts, DExperts, PPLM), the superiority of the proposed framework was verified from three dimensions: overall constraint satisfaction rate, individual constraint satisfaction rate, and collision rate—without sacrificing generation quality.

4. The work establishes an interdisciplinary connection between control theory and NLP, providing a mathematical basis for the predictability of constraint satisfaction in AI systems. From an engineering perspective, the unified constraint framework supports flexible configuration across different scenarios, avoiding the high costs of scenario-specific customization in traditional methods and reducing the deployment threshold in complex environments, thus showing potential for practical implementation.

**Weaknesses:**

1. The H value in MPC-based predictive decoding directly affects performance and computational cost. However, the manuscript fails to provide quantitative data (e.g., inference latency, GPU memory usage) for different H values, making it impossible to determine the method’s applicability in scenarios with strict real-time requirements (such as emergency obstacle avoidance for UAVs). Additionally, no computational optimization strategies were proposed for scenarios with large H values.

2. The manuscript mentions injecting environmental constraints and external knowledge through "hierarchical retrieval-augmented generation or dynamic prompt engineering" but does not specify details of the retrieval mechanism (e.g., type of data source, logic for generating retrieval keywords). Furthermore, no ablation experiments were designed to compare performance with and without external knowledge, making it impossible to quantify the module’s actual contribution to the constraint satisfaction rate and hindering the reproducibility of its implementation details.

3. Ablation experiments for individual modules of LCAD are absent, so the specific contribution of each module to overall performance cannot be determined, and key optimization points remain unclear. In addition, the discussion explicitly identifies the need to balance individual and global constraints in multi-agent scenarios—a critical application direction for LLM agents—but no exploratory work was conducted in this area, which weakens the comprehensiveness of the method.

**Questions:**

1. Provide quantitative data (e.g., inference latency, GPU memory usage) corresponding to different H values in MPC-based predictive decoding. Specify whether there are concrete optimization strategies for the increased computational cost caused by larger H values.

2. Clarify the data source type, retrieval keyword generation logic, and screening criteria for retrieved results of "hierarchical retrieval-augmented generation". Supplement the design of ablation experiments (comparing performance with and without external knowledge) to quantify the module’s gain in constraint satisfaction rate.

3. Conduct ablation experiments by removing MPC-based predictive decoding, multi-dimensional constraint projection, and external knowledge enhancement respectively, to clarify the specific contribution of each individual module to constraint satisfaction rate and collision rate.

---

### Official Review · Reviewer_vPS8 · 2025-11-03

**Soundness:** 2
**Presentation:** 2
**Contribution:** 2
**Rating:** 2
**Confidence:** 4

**Summary:**

This paper introduces the Lyapunov Constraint-Aware Decoding (LCAD) framework, which applies Lyapunov stability theory to the decoding process of large language models (LLMs). The authors argue that existing LLM agents fail to handle multi-constraint environments due to a lack of structured modeling. LCAD maps discrete language generation to continuous constraint optimization by quantifying constraint deviations as Lyapunov functions and integrating predictive decoding, multi-dimensional projection, and environment-aware feedback. Experiments in a simulated UAV wind-farm inspection task show moderate gains in constraint satisfaction and safety over baseline methods such as PPLM and DExperts.

**Strengths:**

1. Creative attempt to connect control theory and LLM decoding, an uncommon and potentially fruitful direction.

2. Provides a unified formalism for handling multiple heterogeneous constraints.

3. Includes an implemented system tested in a realistic UAV simulation, showing measurable though modest improvements.

**Weaknesses:**

1. Theoretical grounding is weak; discrete Lyapunov stability is asserted but not proved.

2. Experimental validation is limited to one domain and lacks statistical or ablation evidence.

3. Heavy mathematical formalism obscures intuition; key mechanisms (e.g., projection and weighting) are difficult to reproduce.

4. Comparative baselines are not clearly adapted to the same constraint-aware setting, making fairness uncertain.

**Questions:**

1. How do you ensure Lyapunov stability in discrete token sequences where gradients are undefined?

2. Could the improvement stem simply from additional constraint-penalty weighting rather than the Lyapunov structure?

3. How sensitive is the framework to hyperparameters such as horizon H and weight $\lambda$?

4. Please include ablations isolating each module.

---

### Note · Authors · 2025-12-01

I have read and agree with the venue's withdrawal policy on behalf of myself and my co-authors.